# M1 Macrophages Promote TRAIL Expression in Adipose Tissue-Derived Stem Cells, Which Suppresses Colitis-Associated Colon Cancer by Increasing Apoptosis of CD133^+^ Cancer Stem Cells and Decreasing M2 Macrophage Population

**DOI:** 10.3390/ijms21113887

**Published:** 2020-05-29

**Authors:** Young Woo Eom, Rokeya Akter, Wanlu Li, Suji Lee, Soonjae Hwang, Jiye Kim, Mee-Yon Cho

**Affiliations:** 1Cell Therapy and Tissue Engineering Center, Yonsei University Wonju College of Medicine, Wonju, Gangwon-do 26426, Korea; yweom@yonsei.ac.kr (Y.W.E.); yoyosong1@naver.com (S.H.); 2Regeneration Medicine Research Center, Yonsei University Wonju College of Medicine, Wonju, Gangwon-do 26426, Korea; 3Department of Pathology, Yonsei University Wonju College of Medicine, Wonju, Gangwon-do 26426, Korea; rokeyahabib94@gmail.com (R.A.); lvvlchina@msn.cn (W.L.); isoojeel@naver.com (S.L.); 4Department of Plastic and Reconstructive Surgery, Yonsei University Wonju College of Medicine, Wonju, Gangwon-do 26426, Korea; gen80@yonsei.ac.kr

**Keywords:** colitis-associated cancer, mesenchymal stem cells, CD133^+^ cancer stem cells, tumor necrosis factor-related apoptosis-inducing ligand (TRAIL), apoptosis

## Abstract

We have previously reported that adipose tissue-derived stem cells (ASCs) cultured at high cell density can induce cancer cell death through the expression of type I interferons and tumor necrosis factor (TNF)-related apoptosis-inducing ligands (TRAIL). Here, we investigated whether TRAIL-expressing ASCs induced by M1 macrophages can alleviate colitis-associated cancer in an azoxymethane (AOM)/dextran sodium sulfate (DSS) animal model. M1 macrophages significantly increased the TRAIL expression in ASCs, which induced the apoptosis of LoVo cells in a TRAIL-dependent manner. However, CD133^knockout^ LoVo cells, generated using the CRISPR-Cas9 gene-editing system, were resistant to TRAIL. In the AOM/DSS-induced colitis-associated cancer model, the intraperitoneal transplantation of TRAIL-expressing ASCs significantly suppressed colon cancer development. Moreover, immunohistochemical staining revealed a low CD133 expression in tumors from the AOM/DSS + ASCs group when compared with tumors from the untreated group. Additionally, the ASC treatment selectively reduced the number of M2 macrophages in tumoral (45.7 ± 4.2) and non-tumoral mucosa (30.3 ± 1.5) in AOM/DSS + ASCs-treated animals relative to those in the untreated group (tumor 71.7 ± 11.2, non-tumor 94.3 ± 12.5; *p* < 0.001). Thus, TRAIL-expressing ASCs are promising agents for anti-tumor therapy, particularly to alleviate colon cancer by inducing the apoptosis of CD133^+^ cancer stem cells and decreasing the M2 macrophage population.

## 1. Introduction

Death receptors (DR) targeted by the tumor necrosis factor α-related apoptosis-inducing ligand (TRAIL) are expressed only in tumor cells and not in normal cells [1,2,3]. TRAIL can regulate CD133^+^ cancer stem cells (CSC) and induce tumor cell-specific apoptosis [4,5]. CD133 is known as a potential CSC marker of tumors in the brain [6,7,8], liver [9], pancreas [10], and colon [11,12]. Furthermore, CD133 is expressed in cells that initiate colorectal cancer [13], and its expression in colorectal cancer cells has been associated with poor prognosis, metastasis, and recurrence [14,15]. Moreover, SNU-475 hepatocellular carcinoma (HCC) cells express high levels of CD133 and are highly susceptible to TRAIL compared to other HCC cells with a low CD133 expression [4]. Additionally, CD133^+^ CSCs of both H460 and H2170 cell lines of non-small cell lung cancer highly express death receptor 5 (DR5) and undergo apoptosis when treated with TRAIL-expressing mesenchymal stem cells (MSCs) [5]. In contrast, Jurkat and breast cancer MCF-7 cells expressing high levels of CD133 are resistant to TRAIL owing to the up-regulation of FLICE-inhibitory protein (FLIP) [16]. Although TRAIL is considered a potent anti-cancer protein [17], it has limited clinical applications, owing to its low stability and short half-life in vivo [3,18]. However, the on-site expression of TRAIL could be an effective anti-tumor treatment. Since MSCs have the characteristics of homing to the tumor site [19,20], MSCs may be used for anti-cancer drug delivery or the local production of anti-tumor therapeutic proteins [21,22]. In fact, the homing property of MSCs has been shown to reduce the side effects of anticancer drugs outside the tumor site, and a small number of MSCs can achieve sufficient therapeutic effects [23]. Therefore, several studies have been conducted to treat tumors using engineered MSC-expressing genes such as *TRAIL* to induce tumor cell-specific apoptosis.

We previously reported that adipose tissue-derived stem cells (ASCs) cultured at a high cell density can induce the death of MCF-7, H460, and Huh7 cells through the expression of type I interferons (IFNs) and TRAIL [24,25,26]. However, in a xenograft tumor model in which human tumor cells were implanted subcutaneously in athymic nude mice with a mutation in the *Foxn1* gene causing a severely compromised immune system, no significant difference in the tumor suppression effect was observed, as was also indicated by the in vitro results [25]. These results suggested that although ASCs express type I IFNs and TRAIL, xenograft tumor models using athymic nude mice have limitations for the evaluation of ASCs anti-tumor effects, maybe because of the lack of immune response in the tumor microenvironment. The tumor microenvironment plays a crucial role in tumor growth; therefore, therapies targeting the cellular components, particularly tumor-associated macrophages, have been actively investigated.

Macrophages are immune cells that can be classified into M1 and M2 types and are interchangeable depending on the immune environment [27]. M1 macrophages typically promote inflammation and monitor immune response, while M2 macrophages mitigate inflammation and promote tumor growth [28]. The expression of CD163, a highly specific marker of M2 macrophages, is associated with tumor proliferation, metastasis, and prognosis [29,30,31]. Recently, Huang et al. introduced a novel therapeutic strategy for non-small cell lung cancer involving TRAIL-functionalized gold nanoparticles that had a selective cytotoxicity to M2-polarized macrophages [32].

Colitis is known to increase the incidence of colorectal cancer; therefore, we investigated whether TRAIL-expressing ASCs could alleviate colitis-associated colon cancer induced in Balb/c wild-type mice by Azoxymethane (AOM)/Dextran Sodium Sulfate (DSS). Overall, our findings support the use of TRAIL-expressing ASCs as a therapeutic approach for colitis-associated colon cancer.

## 2. Results

### 2.1. Enhanced Expression of TRAIL in ASCs Cocultured with M1 Macrophages

The influence of M1 macrophages on the TRAIL expression of ASCs was analyzed by next-generation sequencing (NGS), immunoblotting, and ELISA. The expression of TRAIL mRNA in ASCs cultured at a high density was about 175.51 times higher than that of the control group, and approximately 1597.71 times higher in ASCs co-cultured with M1 macrophages. In other words, the expression of TRAIL mRNA increased 9.1-fold in ASCs co-cultured with M1 macrophages when compared to high-density cultured ASCs. Furthermore, while M1-macrophages did not express TRAIL, macrophages co-cultured with ASCs expressed TRAIL in levels as much as 480.31 times greater than the ones detected for the ASC control group (Figure 1A). Taken together, in macrophages and ASCs co-cultures, TRAIL was expressed by both cells. Still, the TRAIL expression in ASC was about 3.3 times higher than in macrophages, suggesting that ASCs are the major TRAIL source. In addition, the expression of TRAIL protein in cell lysate and conditioned medium (CM) was increased by 5.36 and 2.71 times in ASCs co-cultured with M1 macrophages and high-density cultured ASCs, respectively (Figure 1B). Moreover, the concentrations of the secreted TRAIL in ASCs cultured at a high density and co-cultured with M1 macrophages were 135.37 ± 12.76 and 475.22 ± 18.55 pg/mL, respectively (Figure 1C). These results suggest that M1 macrophages significantly increased the expression of TRAIL in ASCs.

### 2.2. Toxicity of TRAIL and CM in LoVo Cells

To evaluate the TRAIL-dependent toxicity in colon cancer cells (LoVo), we used CM obtained from high-density cultures of ASCs. TRAIL (100 ng/mL) reduced the viability of LoVo cells in a time-dependent manner (Figure 2A). Similarly, CM increased the LoVo toxicity by about 13.17% (Figure 2B). Moreover, when the TRAIL in CM was neutralized with an anti-TRAIL antibody, the toxicity of CM in the LoVo cells was reduced (Figure 2C). These results suggest that the toxicity in LoVo cells by CM was TRAIL-dependent. In addition, in LoVo cells co-cultured indirectly with ASCs, the Annexin +/7-Aminodactinomycin (7-AAD) + population (apoptotic cells) was 44.2 ± 7.8%, indicating that apoptosis was increased by 1.7 times as compared to that in the control group (26.0 ± 4.2%) (Figure 2D).

### 2.3. TRAIL Resistance of LoVo-CD133 KO Cells

We investigated the TRAIL sensitivity of CD133^+^ CSCs by comparing the TRAIL effects on CD133^knockout^ LoVo cells (LoVo-CD133 KO). We generated the LoVo-CD133 KO cells (Figure 3A) using the CRISPR-Cas9 gene-editing system, as in our previous report [33]. Interestingly, the TRAIL-induced apoptosis was dramatically increased in the CD133^+^ LoVo cells, while there was no significant difference in the apoptosis of LoVo-CD133 KO cells (Figure 3B,C). In addition, the CM obtained from ASCs cultured at high density increased the apoptosis of LoVo cells, while no apparent effect was detected for LoVo-CD133 KO cells (Figure 3C).

### 2.4. Suppression of Colon Cancer Development and CD133 Expression by TRAIL-Expressing ASCs

Next, we investigated whether TRAIL-expressing ASCs could alleviate cancer development in AOM/DSS-induced colon cancer model. AOM was injected once into the intraperitoneal cavity on d 0, DSS was administered three times a week with drinking water, and TRAIL-expressing ASCs were transplanted i.p. 1 d before the DSS administration (Figure 4A). After 56 d, the colon length was found to be reduced in the mice from the AOM/DSS group (Figure 4B,D), and an average of 12.8 colon tumors per mouse was observed (Figure 4C). Similarly, mice from the AOM/DSS + ASCs group had a reduced colon length (Figure 4B,D); however, the average number of tumors was 1.4 per mouse, which was significantly less than that of mice from the AOM/DSS group (Figure 4C). Histologically, all the tumors induced by AOM/DSS were well to moderately differentiated adenocarcinomas (Figure 5) and were variable in size, but AOM/DSS + ASC-treated mice exhibited tumors less than 2 mm in size. The non-tumor colonic mucosa showed multifocal scar alterations in the AOM/DSS and AOM/DSS + ASCs groups, but there was no significant active inflammation in any group.

Immunohistochemical (IHC) staining revealed CD133 expression in the luminal side of most tumor glands in AOM/DSS-induced colon cancer (Figure 5H) but not in normal mucosa (Figure 5G). In contrast, the AOM/DSS + ASCs group revealed focal CD133 expression in few tumor glands (Figure 5I).

### 2.5. Decreasing M2 Macrophage Population Using TRAIL-Expressing ASCs

IHC staining for CD163, a specific marker for M2 macrophages, was diffusely scattered in the lamina propria of mucosa in all groups, including the control (Figure 6A–C). In the AOM/DSS group, the number of CD163-positive M2 macrophages in the tumoral area was significantly increased (71.7 ± 11.2) in contrast to that of the control group (12.7 ± 2.5; *p* < 0.001). The AOM/DSS + ASCs group had a significantly lower number of CD163-positive M2 macrophages (45.7 ± 4.2) than that of the AOM/DSS group (Figure 6G,H; *p* <0.001). Moreover, non-tumoral mucosa showed similar findings to the tumoral area (AOM/DSS (94.3 ± 12.5) vs. AOM/DSS + ASCs (30.3 ± 1.5), *p* < 0.001). For F4/80 immunostaining, there was a higher number of F4/80-positive cells than CD163-positive macrophages in the control samples (Figure 6D,G; CD163, 12.7 ± 2.1 vs. F4/80, 35 ± 5). Further, mucosa from the AOM/DSS and AOM/DSS + ASCs groups showed intratumoral, peritumoral, and non-tumoral mucosa-infiltrating macrophages (Figure 6E,F). The ASC treatment also lowered the number of F4/80-positive macrophages to 92.3 ± 5.9 in tumor and 103.3 ± 6.1 in non-tumoral mucosa compared with 105.0 ± 8.2 in tumor and 103.3 ± 6.1 in non-tumoral mucosa following the AOM/DSS treatment, but the difference was not statistically significant (*p* > 0.05). These results suggest that TRAIL-expressing ASCs can suppress tumor development by selectively reducing the number of M2 macrophages during colitis-associated colon cancer development.

## 3. Discussion

In this study, we observed that TRAIL expression in ASCs was significantly increased by M1 macrophages, and TRAIL-expressing ASCs could effectively alleviate colon cancer by suppressing CD133^+^ CSCs and M2 macrophages in the tumor microenvironment. TRAIL can selectively kill only cancer cells; therefore, studies using TRAIL to treat cancer have been actively conducted [4,5]. Based on the homing characteristic of MSCs into the site of inflammation or tumor, MSCs are used to treat cancers as drug and therapeutic gene carriers [5,21,34,35]. In particular, MSCs engineered with the TRAIL gene have demonstrated therapeutic effects in several tumor models [5,21,35]. We reported for the first time that ASCs co-cultured with M1 macrophages secreted high levels of TRAIL without gene manipulation. However, in this study the CM from ASCs cultured at a high density were used to evaluate the effect of TRAIL, excluding the anti-tumor activity of M1 macrophages. Although the cytotoxicity of CM in LoVo cells was about 13%, the ASC transplantation in the AOM/DSS tumor model reduced tumor development by about 90%. The difference in anti-cancer efficiency in vitro and in vivo could be explained by the presence or absence of M1 macrophages and/or expression level of TRAIL. That is, in vivo M1 macrophages might increase the TRAIL expression in ASCs and subsequently suppress the development and progression of colon cancer.

Furthermore, AOM/DSS can initiate and promote relatively strong and reproducible colitis-associated cancer through repeated cycles of colitis caused by DSS following DNA damage by AOM [36,37,38,39,40]. DSS-induced colitis induces the massive infiltration of T and B lymphocytes, neutrophils, granulocytes, and macrophages, and they express various pro-inflammatory cytokines, including tumor necrosis factor (TNF)-α; interleukin (IL)-6, IL-8, IL-12, and IL-17; and IFN-γ [41,42]. IFN-γ plays an important role in the initiation of DSS-induced colitis by increasing the expression of three chemokines: monokines induced by IFN-γ (MIG), IFN-inducible protein 10 (IP-10), and monocyte chemoattractant protein-1 (MCP-1) [43]. In addition, IFN-γ activates M1 macrophages, which then produce TNF-α and reactive oxygen species (ROS) and respond to the pathogenesis of colitis [44,45]. M1 macrophages are potent scavengers of invading pathogens which activate the innate immune system and then induce the activity of the acquired immune system [46]. However, in the AOM/DSS tumor model, M2 macrophages have been shown to initiate, promote, and induce migration of the colon cancer [29]. In other words, M1-to-M2 polarization occurs during colon cancer development and metastasis, and M2 macrophages exhibit a pro-tumor role [47]. In our previous report, in the DSS colitis model the M1 macrophage population increased significantly, while ASC treatment decreased the M1 population but did not increase the M2 population [45]. In this study, M2 macrophages were increased in the tumor and non-tumoral mucosa of mice from the AOM/DSS group and remarkably reduced in the AOM/DSS + ASCs group. M2 macrophages are known to have a higher sensitivity to TRAIL than M1 macrophages do [32]. Therefore, in the AOM/DSS model, TRAIL-expressing ASCs exposed to M1 macrophages were expected to lower the colon cancer development by controlling the pro-tumor M2 macrophage and CD133^+^ CSC populations.

Another important characteristic of MSCs is to modulate the activity of various immune cells. MSCs express various kinds of immunomodulators—including indoleamine 2,3-dioxygenase, prostaglandin E2, TNF-α-stimulated protein/gene 6, NO, IL-6, IL-10, and HLA-G—to inhibit the activity of T- and B-cells, macrophages, and natural killer cells [48,49]. Chronic inflammation increases the risk of cancer development, promotes tumor progression, and increases metastasis [50,51,52]. In the early phase of tumor development, several cytokines and ROS secreted by tumor-infiltrating immune cells induce epigenetic changes in premalignant lesions and inhibit tumor suppressor genes [53]. Moreover, during tumor promotion, immune cells produce several cytokines and chemokines essential for tumor cell survival and proliferation, resulting in tumor progression and metastasis [54]. Therefore, modulating the activities of immune cells through MSCs may be a mechanism for reducing tumor development and progression. However, according to our previous results, the tumor-suppressive effect of ASCs cultured at a high density in the xenograft tumor model using athymic nude mice was not significant [25]. Moreover, although it has been reported that the function of macrophages and DCs is maintained in xenograft tumor models using athymic nude mice [55], it remains controversial whether the distribution and function of macrophages and DCs in the tumor microenvironment is normal. However, the current study demonstrated that in the colitis-associated cancer model, ASCs modulated the macrophage population during colon cancer development.

In summary, TRAIL-expressing ASCs were able to alleviate the development and progression of colon cancer by inducing the apoptosis of CD133^+^ CSCs and reducing the number of M2 macrophages. To understand the therapeutic mechanisms of MSCs in an in vivo system, future studies analyzing differences in immune cell populations in the tumor microenvironment after MSC transplantation are warranted. Moreover, to optimize the effectiveness of the treatment, it is necessary to determine at which stage of the tumor development MSCs must be transplanted. These findings would help advance the treatment of colon cancer using MSCs.

## 4. Materials and Methods

### 4.1. Cell Culture

This study was approved by the Institutional Review Board of the Yonsei University Wonju College of Medicine (IRB n° 2011–58, approval date: 22 December 2013) and the scientific use of human adipose tissues was permitted by written informed consent from three healthy donors (24–38 years of age). ASCs were isolated using a modified protocol [24,56] and sub-cultured with low-glucose Dulbecco’s modified Eagle’s medium (DMEM; Gibco, Rockville, MD, USA) supplemented with 10% fetal bovine serum (FBS, Gibco) and penicillin/streptomycin (Gibco). For the experiments, ASCs were seeded at 4 × 10^4^ cells/cm^2^. After the 3-day culture period, the CM was collected, filtered with syringe-driven filters (0.45 μM), and stored at −80 °C until further use.

The LoVo cells were maintained in Roswell Park Memorial Institute-1640 medium (RPMI-1640) (Hyclone, Logan, UT, USA) supplemented with 10% FBS (Gibco) and penicillin/streptomycin (Gibco). The CD133^knockout^ LoVo cells (LoVo-CD133 KO) were generated using the CRISPR-Cas9 gene editing system, as described in our previous report [34]. For the indirect co-culture of ASCs and LoVo cells, a transwell plate (Costar, Kennebunk, ME, USA) was used. The ASCs were cultured for three days in the upper chamber without LoVo cells in the lower chamber, the LoVo cells were cultured in the lower chamber for one day without ASCs in the upper chamber, and then the upper and lower chambers were assembled to initiate the co-culture.

The human monocytic cell line THP-1 was maintained in complete RPMI-1640 (Gibco; supplemented with 10% FBS, penicillin/streptomycin, and 2 mM L-glutamine). Macrophage differentiation (from THP-1 cells) was induced with 100 nM of phorbol ester 12-O-tetradecanoylphorbol-13-acetate (TPA, Sigma, San Diego, CA, USA) for 2 days. The macrophages were co-cultured with ASCs (in trans-well plates; Corning, Lowell, MA, USA) under treatment with 20 ng/mL of IFN-γ (R&D Systems, Minneapolis, MN, USA) and 10 pg/mL of lipopolysaccharide (LPS, Sigma). After indirect co-cultures, the total mRNA and proteins were recovered separately from the macrophages and ASCs.

### 4.2. Next-Generation Sequencing (NGS)

The total RNA was extracted from 1 × 10^5^ cells using TRIzol Reagent (Gibco BRL) according to the manufacturer’s instructions. The libraries were prepared for 150 bp paired-end sequencing using TruSeq Stranded mRNA Sample Prep Kit (Illumina, San Diego, CA, USA). The mRNA samples were purified and fragmented from 1 μg of total RNA using oligo (dT) magnetic beads, and the fragmented mRNAs were synthesized as single-stranded cDNAs through random hexamer priming. These were used as templates for second-strand synthesis and the preparation of double-stranded cDNA. After sequential end repairing, A-tailing, and adapter ligation, cDNA libraries were amplified by polymerase chain reaction (PCR). The quality of these cDNA libraries was evaluated with the Agilent 2100 BioAnalyzer (Agilent, Santa Clara, CA, USA), and the cDNA libraries were quantified using the KAPA library quantification kit (Kapa Biosystems, Wilmington, MA, USA) according to the manufacturer’s protocol. Following the cluster amplification of denatured templates, sequencing was performed of the paired-end reads (2 × 150 bp) using Illumina NovaSeq6000 (Illumina, San Diego, CA, USA).

### 4.3. Immunoblotting

The proteins were separately prepared from the cells or CM. The cells were lysed in a sodium dodecyl sulfate-polyacrylamide gel electrophoresis (SDS–PAGE) sample buffer (62.5 mM Tris-HCl (pH 6.8), 1% SDS, 10% glycerol, and 5% β-mercaptoethanol) and the CM was mixed (1:1) with 2× sample buffer. The proteins were boiled for 5 min, subjected to SDS–PAGE, and transferred to an Immobilon polyvinylidene difluoride (PVDF) membrane (Millipore, Burlington, MA, USA). The membrane was blocked with 5% skim milk in Tris-HCl buffered saline containing 0.1% Tween 20 and then incubated with primary antibodies against TRAIL (1:1000, R&D Systems, Minneapolis, MN, USA), β-actin (1:1000, Santa Cruz Biotech, Santa Cruz, CA, USA), and CD133 (1:1000, Miltenyi Biotec, Bergisch Gladbach, Germany), followed by peroxidase-conjugated secondary antibodies (1:2000, Santa Cruz Biotech). Then, the membrane was treated with EZ-Western Lumi Pico (DOGEN, Seoul, Korea) and visualized using the ChemiDoc XRS+ system (Bio-Rad, Hercules, CA, USA).

### 4.4. Enzyme-Linked Immunosorbent Assay (ELISA)

The CM was recovered from ASCs cultured at a high density for 3 days, and the concentration of secreted TRAIL (sTRAIL) was measured by the human TRAIL Quantikine ELISA kit (R&D Systems) according to the manufacturer’s instructions.

### 4.5. Cytotoxicity Assay

The LoVo cells were seeded at a density of 1 × 10^4^ cells/cm^2^ in 96-well plates and cultured for 24 h. The cells were treated with human recombinant TRAIL (R&D systems), incubated for 24 h, treated with 10 μL of water soluble tetrazolium salt (WST)-1 reagent (Roche, Indianapolis, IN, USA), and incubated for 1 h at standard culture conditions. Then, the absorbance was measured using a microplate reader (Molecular Devices, San Jose CA, USA) at 450 nm.

### 4.6. Apoptosis Assay

The PE-Annexin-V apoptosis detection kit I (BD Biosciences, San Diego, CA, USA) was used according to the manufacturer’s instructions. The cells were harvested, washed twice with cold PBS, and re-suspended in a binding buffer. The cells were stained with PE-Annexin-V and 7-aminoactinomycin D (7-AAD) for 15 min at room temperature in the dark and then analyzed without washing using a flow cytometer (FACSAria III, BD Biosciences) within 1 h after the staining.

### 4.7. Animal Study

Six-week-old Balb/c mice (male, 18–22 g) were purchased from Orient Bio, Inc. (Seongnam, Korea). The mice were maintained in a 12-h light/12-h-dark cycle at 23 °C. All the animal care and experiments were conducted in accordance with the Guide for Animal Experiments published by the Korea Academy of Medical Sciences and approved by the Institutional Animal Care and Use Committee of Yonsei University, Wonju College of Medicine (YWC-180117-1, approval date: 23 September 2018). The mice were divided into the following 3 groups (*n* = 5 per group): AOM/DSS/ASCs-untreated control, AOM/DSS, and AOM/DSS + ASCs. To prepare the AOM/DSS model of colon cancer [35], the AOM (10 mg/kg, Sigma) was injected intraperitoneally (i.p.) and 7 d later mice were administered drinking water with 1.5% DSS (MP Biomedicals, Santa Ana, CA, USA) for 7 d, followed by normal water without DSS for 14 d. This DSS administration cycle was repeated twice and ASCs (1 × 10^6^ cells/mouse) were injected i.p. 3 times, on day 6, 27, and 48. The mice were sacrificed on day 56 to analyze the colon tumor development.

### 4.8. Immunohistochemical (IHC) Staining

IHC staining of the paraffin-embedded tissue sections was performed as described in our previous report [57]. The slides were incubated with the monoclonal antibodies against CD133 (Miltenyi Biotec) for CSCs, CD163 (Abcam) for M2 macrophages, and F4/80 (cell signaling technology) for pan macrophages for 2 h at 37 °C in an autostainer using an Ultra View Universal DAB Detection Kit (Benchmark XT, Ventana Medical Systems, Tucson, AZ, USA). Then, the slides were analyzed using an Olympus BX51 microscope (Olympus, Tokyo, Japan), and the positive staining of CD163 and F4/80 was scored based on the number of positive cells/high power field (HPF, ×400) for comparison.

### 4.9. Statistical Analysis

Data are presented as the mean ± standard error of the mean. To compare the group means, Student’s *t*-test and one-way analysis of variance were used, followed by Scheffe’s test. A *p*-value of <0.05 was considered statistically significant.

## 5. Conclusions

M1 macrophages significantly increased the TRAIL expression in ASCs and subsequently induced the apoptosis of colon cancer LoVo cells. This TRAIL-dependent response alleviated the development and progression of colon cancer, particularly by inducing the apoptosis of CD133^+^ CSCs and reducing the M2 macrophage population. However, CD133^knockout^ LoVo cells were resistant to TRAIL. These results suggest that CD133^+^ CSCs are potential targets of the TRAIL-expressing ASCs for treating colitis-associated colon cancer. In conclusion, for chronic inflammatory diseases, including colitis ASCs can be used as therapeutic agents to slow the onset and progression of tumors and relieve inflammation through TRAIL expression.

## Figures and Tables

**Figure 1 ijms-21-03887-f001:**
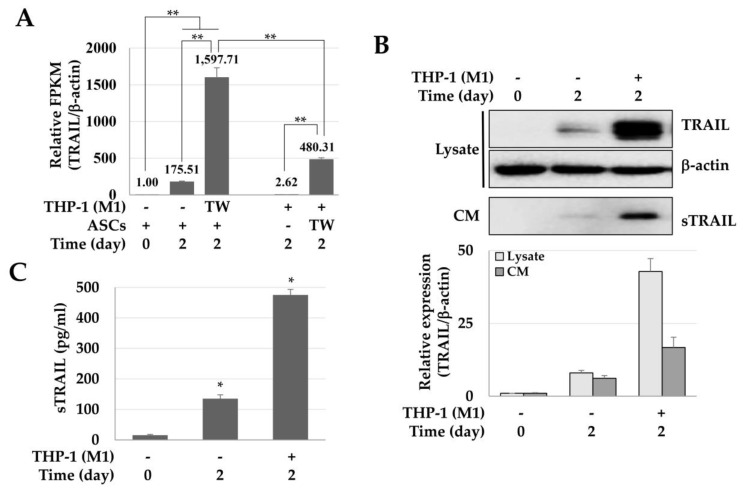
Enhanced expression of tumor necrosis factor (TNF)-related apoptosis-inducing ligand (TRAIL) in adipose tissue-derived stem cells (ASCs) co-cultured with M1 macrophages. ASCs were cultured at high-density or co-cultured with M1 macrophages (THP-1) for 2 days and harvested to analyze the TRAIL mRNA and protein, and conditioned media were stored after centrifugation to detect the secreted TRAIL (sTRAIL). (**A**) Relative expression of TRAIL in ASCs cultured at a high density or with M1 macrophages. The relative expression of TRAIL was analyzed by a next-generation sequencing assay, and the relative values are presented in Fragments Per Kilobase Million (FPKM). ** *p* < 0.01. (**B**) Expression of TRAIL in cell lysates and conditioned medium (CM). TRAIL was detected in cell lysates and CM using immunoblotting. β-actin was used for normalization. (**C**) Expression of sTRAIL. * *p* < 0.05.

**Figure 2 ijms-21-03887-f002:**
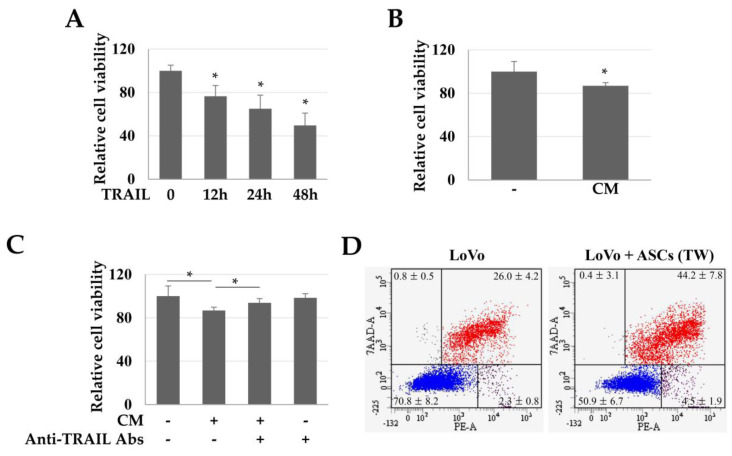
Toxicity of TRAIL and conditioned medium (CM) in LoVo cells. LoVo cells were treated with human recombinant TRAIL or CM and indirectly co-cultured with ASCs. (**A**) Toxicity of LoVo cells by TRAIL at different times. Cell viability decreased in a time-dependent manner. Data are presented as the mean ± SE of three independent experiments. ** p* < 0.05. (**B**) Relative viability of LoVo cells. LoVo cells were treated with CM for 24 h and toxicity was detected using a WST-1 reagent. * *p* < 0.05. (**C**) TRAIL-dependent toxicity of CM in LoVo cells. To detect whether the toxicity of CM in LoVo cells was TRAIL-dependent, the TRAIL protein in CM was neutralized with anti-TRAIL antibodies and the cytotoxicity was evaluated using a WST-1 reagent. ** p* < 0.05. (**D**) ASC-induced apoptosis of LoVo cells. To investigate the direct effects of ASCs on the apoptosis of LoVo cells, the LoVo cells were co-cultured with ASCs and then analyzed by Annexin-V/7-AAD staining and flow cytometry.

**Figure 3 ijms-21-03887-f003:**
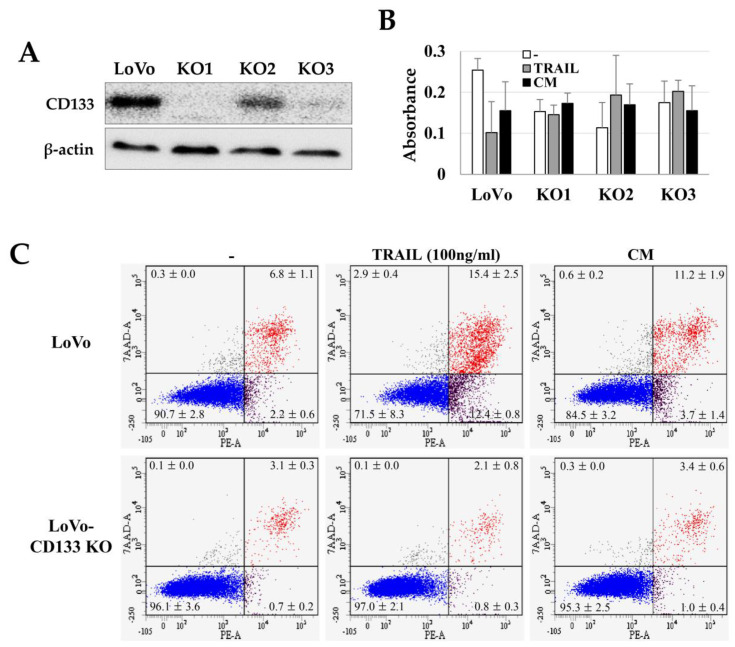
TRAIL sensitivity in LoVo and CD133^knockout^ LoVo cells (LoVo-CD133 KO cells). Three independent LoVo-CD133 KO cells (KO1, KO2, KO3) were generated using the CRISPR-Cas9 gene editing system and culturing a single colony. (**A**) Immunoblotting of CD133 expression in LoVo-CD133 KO cells. (**B**) Toxicity of TRAIL in LoVo-CD133 KO cells. TRAIL reduced the viability of LoVo cells, but TRAIL-dependent toxicity was not observed in all the KO cells. Data are presented as the mean ± SE of three independent experiments. (**C**) Sensitivity to TRAIL by LoVo and LoVo-CD133 KO1 cells. TRAIL-induced apoptosis was analyzed by Annexin-V/7-AAD staining and flow cytometry.

**Figure 4 ijms-21-03887-f004:**
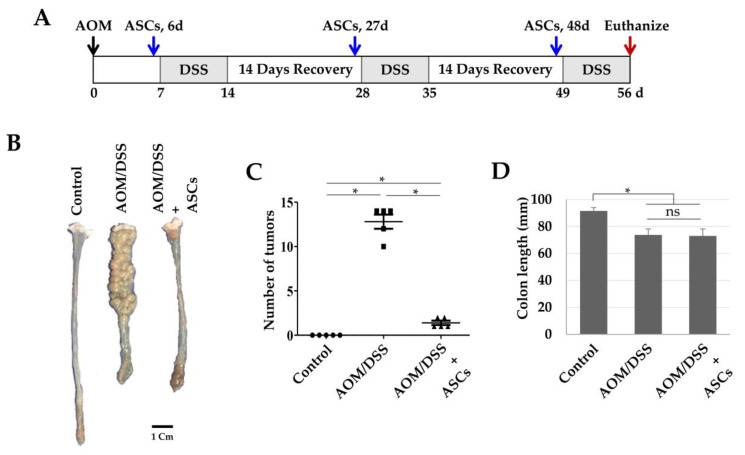
Suppression of colon cancer development by ASCs. (**A**) Schematic timeline for azoxymethane (AOM)/dextran sodium sulfate (DSS)-induced colon cancer and ASC transplantation. (**B**) Gross tumor burden by AOM/DSS and inhibition of tumor development by ASCs. (**C**) Reduced tumor numbers by ASC transplantation. The number of tumors in colons obtained from the control, AOM/DSS, and AOM/DSS + ASCs groups (*n* = 5) were counted. * *p* < 0.05. (**D**) Lengths of colons obtained from the control, AOM/DSS, and AOM/DSS + ASCs groups. * *p* < 0.05. ns: not statistically significant.

**Figure 5 ijms-21-03887-f005:**
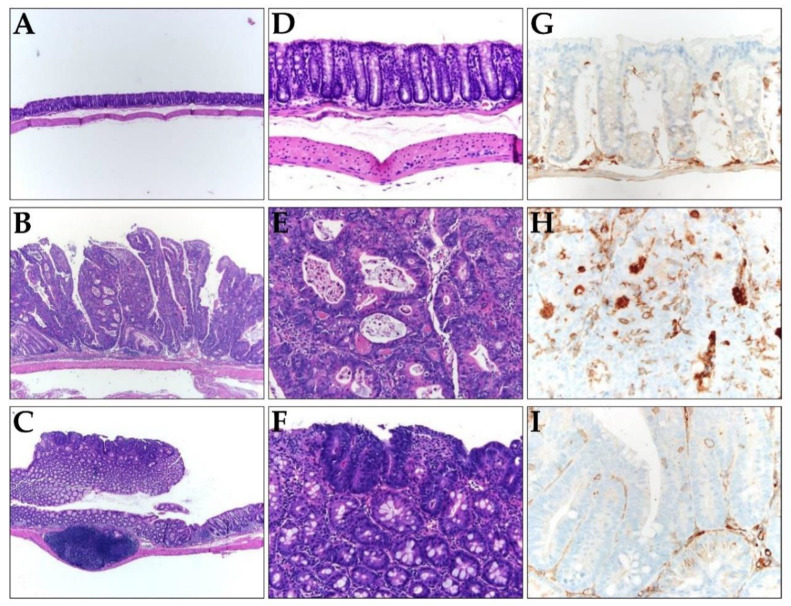
(**A**–**F**) Representative histology images of colons resected from (**A**,**D**) the control, (**B**,**E**) AOM/DSS, and (**C**,**F**) AOM/DSS + ASCs groups. (**A**–**C**) Low-power view (×100) of colon tissues stained with hematoxylin and eosin (**H**,**E**), showing most tumors are bigger in mice from (**B**) the AOM/DSS group compared to the small and superficial tumors in mice from (**C**) the AOM/DSS + ASCs group. (**D**–**F**) Higher magnification (×200) of H&E staining, showing (**D**) normal mucosa, (**E**) AOM/DSS-induced moderately differentiated adenocarcinoma, and (**F**) the tumor in AOM/DSS + ASCs group, which is similar to that of the AOM/DSS group. (**G**–**I**) The immunohistochemical staining of CD133 in (**G**) the control, (**H**) AOM/DSS, and (**I**) AOM/DSS + ASCs (×400). (**G**) No CD133 expression in the normal mucosa. (**H**) Strong expression of CD133 along the lumen structure of the AOM/DSS-induced tumor glands. (**I**) Very rare and focal CD133 positivity in the tumor from the AOM/DSS + ASCs group.

**Figure 6 ijms-21-03887-f006:**
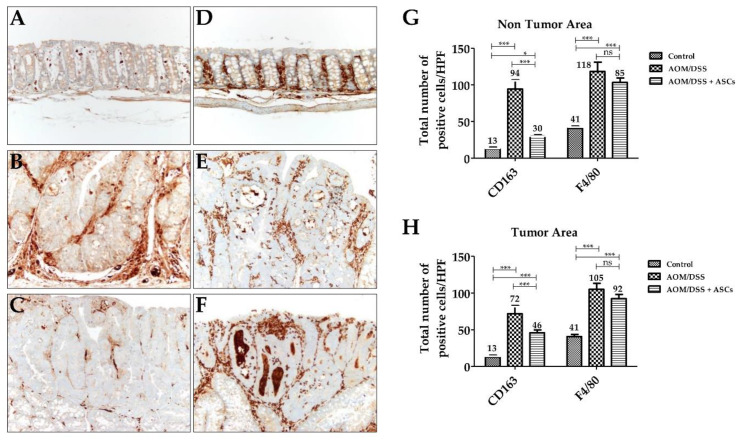
Changes in macrophages by TRAIL-expressing ASCs. (**A**–**C**) CD163 immunostaining (×200). (**A**) Normal mucosa with a few scattered CD163-positive M2 macrophages in lamina propria. (**B**) AOM/DSS-induced tumor with numerous M2 macrophages in the peritumoral area. (**C**) AOM/DSS + ASC-treated group showing a reduced number of M2 macrophages. (**D**–**F**) F4/80 immunostaining (×200). (**D**) Normal mucosa. Tumors from (**E**) AOM-DSS group and (**F**) AOM/DSS + ASCs group showing numerous macrophages. (**G**,**H**) Comparison of the number of macrophages between the three groups. There was a significantly higher number of F4/80-positive M2 macrophages in AOM/DSS-induced tumoral (**H**) and non-tumoral (**G**) mucosa than in the CD163-positive M2 macrophages. ASCs treatment significantly reduced the number of CD163-positive M2 macrophages, but not F4/80-positive M2 macrophages in the tumor and non-tumor areas. (* *p* < 0.05, *** *p* < 0.001, ns *p* > 0.05). ns: not statistically significant.

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
