# Peer review of "M1 Macrophages Promote TRAIL Expression in Adipose Tissue-Derived Stem Cells, Which Suppresses Colitis-Associated Colon Cancer by Increasing Apoptosis of CD133+ Cancer Stem Cells and Decreasing M2 Macrophage Population"

_ijms, 2020, doi:10.3390/ijms21113887_

Round 1

Reviewer 1 Report

  1. Please write full name of ASCs in the title.
  2. At Line 16, the full name of ASCs is different from that at Line 56.
  3. At Line 79, two results present.
  4. Please write full name of CM at Line 86.
  5. Please provide IRB approval number.
  6. To suggest authors: Primary cells should be added and performed the same study in the Figure 1.
  7. Line 280: ref 34 format should be corrected.
  8. Since TRAIL-induced apoptosis was dramatically increased in CD133+ LoVo cells, while there was no significant difference in apoptosis of LoVo-CD133 KO cells, please add a colon cancer cell line to performed the same study of Lovo cell line.
  9. Please provide the high qualitive Fig. 2D and Fig. 3C.
  10. Please provide whether the significant difference in the Fig. 4C-D.

Author Response

We would like to thank Reviewer 1 and 2 for the valuable comments, which have helped us to improve this manuscript.

In response to the comment that "English language and style are fine/minor spell check required", English proofreading was performed again through a High-quality English editing services from editage Co. (https://www.editage.co.kr), and the newly revised parts were changed using the “Track Changes” option of Microsoft Word program. We have revised and updated the manuscript to answer to all the issues raised, and hope we have improved the manuscript sufficiently for publication in the International Journal of Molecular Sciences. We provide below a point-by-point response to each of the comments.

Reviewer 1;

First of all, we would like to thank the Reviewer 1 for his/her comments, which we believe helped us to improve our manuscript.

  1. Please write full name of ASCs in the title.

Response: According to the reviewer's comments, we modified the ASCs in the title to adipose tissue-derived stem cells.

  1. At Line 16, the full name of ASCs is different from that at Line 56.

Response: Thank you for the constructive comment. The full name of ASCs at Line 56 of original manuscript was changed to adipose tissue-derived stem cells.

  1. At Line 79, two results present.

Response: We apologize for using the same word twice. As the reviewer pointed out, we deleted one Results word.

  1. Please write full name of CM at Line 86.

Response: Thank you for the constructive comment. In the line 86 of original manuscript, the full name of CM was changed to conditioned medium (CM) and the conditioned medium (CM) in line 275 has been modified to CM.

  1. Please provide IRB approval number.

Response: IRB number (2011-58) was already presented in the 5.1. cell culture section (line 270).

  1. To suggest authors: Primary cells should be added and performed the same study in the Figure 1.

Response: In this reviewer's comment, we consider primary cells to mean adipose tissue-derived stem cells (ASCs) that were treated nothing. Their results are the first bar graphs (Fig. 1A, 1B, and 1C) and the band of western blotting (Fig. 1B).

  1. Line 280: ref 34 format should be corrected.

Response: Thank you for the constructive comment. We corrected ref 34 format in line 280 to [34].

  1. Since TRAIL-induced apoptosis was dramatically increased in CD133+ LoVo cells, while there was no significant difference in apoptosis of LoVo-CD133 KO cells, please add a colon cancer cell line to performed the same study of Lovo cell line.

Response: We greatly appreciate this important point of the reviewer.

The aims of this study were to identify the CD133 expression-dependent TRAIL sensitivity of the colorectal cancer cell line, and further to evaluate the effect of inhibiting the development and progression of colon cancer by TRAIL-expressing ASCs in the animal model. We confirmed the cytotoxicity of TRAIL in other colon cancer cell lines DLD-1, SNU-407, and Sw-480 in addition to LoVo cells, and observed that DLD-1 cells are more sensitive to TRAIL than LoVo cells. Similar to LoVo cells, ASC also induced apoptosis of DLD-1 cells.

Unfortunately, due to several problems, we were unable to make CD133knockout DLD-1 cells using the CRISPR-Cas9 gene editing system, and CD133 expression-dependent TRAIL toxicity was not confirmed in DLD-1 cells. Therefore, although we observed that DLD-1 cells induce cytotoxicity by TRAIL or TRAIL-expressing ASCs, it is not appropriate to include the relevant results in this article because these cytotoxicities do not meet our research objectives. Since we have not evaluated whether TRAIL induced apoptosis of CD133+ population in the colon cancer cell lines DLD-1, SNU-407 and Sw480 other than LoVo cells, we think it doesn't help our readers understand our conclusion that TRAIL modulates CD133+ cells to relieve colon cancer.

However, to answer the reviewer's comments, we add the relevant results only to the revision note, not the revised MS.

  1. Please provide the high qualitive Fig. 2D and Fig. 3C.

Response: We greatly appreciate this important point raised by the reviewer. Following the reviewer's suggestion, we re-analyzed the results of our apoptosis assays, displaying each population in a different color. We further increased the resolution and the size of Figures 2D and 3C.

  1. Please provide whether the significant difference in the Fig. 4C-D.

Response: We greatly appreciate this important point of the reviewer. Significant differences in Figure 4C and 4D are now shown in the revised manuscript.

Reviewer 2 Report

In the current manuscript, the authors report that the secretion of TRAIL from adipose tissue-derived stem cells induced by M1 macrophages could suppress the development of colon cancer. TRAIL promoted the apoptosis of cancer cells in vitro, and inoculation with ASCs leads to less tumor formation, decreased CD133+ tumor cells as well as M2 polarized macrophages. The study is simple, and the finding could be of significant importance if being translated into clinical therapy. However, the study is flawed by lacking proper controls and missing key evidence. In general, although the storyline is there, the connection between each conclusion is missing. Several major concerns are listed below.

Major comments

1 the major source of TRAIL in macrophages cocultured ASCs is not clear. The authors did not rule out the possibility that increased TRAIL may be secreted by the macrophages instead of ASCs in the coculture system. A detailed protocol should be provided regarding how to avoid the contamination of macrophages derived materials (such as mRNA or proteins) in the interpretation of increased TRAIL during coculture.

2 the experimental design in figure 2 is not rigorous and compromised by the possibility that other components than TRAIL in the CM may modulate the corresponding phenotypes on tumor cells. A good design is using TRAIL KO ASCs as a control group side by side with the CM from the WT ASCs. Meantime, the reviewer is curious about the CM used in figure 2. The authors claimed that the CM was derived from high-density cultures of ASCs, which may be greatly different with that from macrophages cocultured ASCs. If the authors still want to claim the role of M1 macrophages in inducing TRAIL and the consequence on the tumor cells, CMs from M1 macrophages cocultured WT and TRAIL KO ASCs should be also investigated.

3 the function of TRAIL in CD133+ tumor cells is not novel. Therefore, to further support the authors’ claim in Figure 3, CMs from both TRAIL expressing ASCs should be tested in both WT and CD133 KO LoVo cells.

4 Again, TRAIL KO ASCs should also be studied in vivo to further support their claims in Figure 4-6. Also, I do not see any evidence here regarding the in vivo role of M1 macrophages. Do the authors just want to simply show that M1 macrophages promote TRAIL secretion from ASCs in vitro?

Minor comments

1 It is surprising that CD133 KO LOVO cells exhibit slightly reduced apoptosis compared to the control cells in Figure 3C, which contradicts the tumor-promoting role of CD133. The authors should re-check this data. In addition, Figure 3B is kind of confusing to me. Do the authors suggest that CD133 KO does not affect the cell viability of LOVO cells?

2 the authors should describe how they prepare M1 THP-1 macrophages. This is important to interpret the data regarding the specific role of M1 macrophages in Figure 1. It is also interesting to test if M2 macrophages show similar or different behavior in promoting the secretion of TRAIL from ASCs.  

Author Response

We would like to thank Reviewer 1 and 2 for the valuable comments, which have helped us to improve this manuscript.

In response to the comment that "English language and style are fine/minor spell check required", English proofreading was performed again through a High-quality English editing services from editage Co. (https://www.editage.co.kr), and the newly revised parts were changed using the “Track Changes” option of Microsoft Word program. We have revised and updated the manuscript to answer to all the issues raised, and hope we have improved the manuscript sufficiently for publication in the International Journal of Molecular Sciences. We provide below a point-by-point response to each of the comments.

Reviewer 2

First of all, we would like to thank the Reviewer 2 for his/her comments, which helped us to improve this manuscript. Furthermore, we apologize in advance for not being able to faithfully carry out all the experiments the reviewer has requested. Please note that the journal requested us to deliver this revision within 10 days, which limited any complex experimental procedures. However, we did our best to respond to your comments, and we performed all the experiments we could, considering the time constraints.

Major comments

1 the major source of TRAIL in macrophages cocultured ASCs is not clear. The authors did not rule out the possibility that increased TRAIL may be secreted by the macrophages instead of ASCs in the coculture system. A detailed protocol should be provided regarding how to avoid the contamination of macrophages derived materials (such as mRNA or proteins) in the interpretation of increased TRAIL during coculture.

Response: We greatly appreciate this critical comment from the reviewer.

To respond to the comment that the main source of TRAIL in our co-culture system may not be ASCs, we further added the next generation sequencing (NGS) results on TRAIL expression in M1-macrophages alone co-cultured with ASCs to Figure 1A. In the added results, M1-macrophages did not express TRAIL. However, macrophages co-cultured with ASCs expressed TRAIL about 480.31 times higher than the levels detected for the ASC control group. Taken together, in macrophages and ASCs co-cultures, TRAIL was expressed by both cells. Still, TRAIL expression in ASC was about 3.3 times higher than in macrophages, suggesting that ASCs are indeed the major TRAIL source. These results were added to section 2.1 of the revised manuscript as following (please see lines 93-97):

“Furthermore, while M1-macrophages did not express TRAIL, macrophages co-cultured with ASCs expressed TRAIL in levels as much as 480.31 times than the ones detected for the ASC control group. Taken together, in macrophages and ASCs co-cultures, TRAIL was expressed by both cells. Still, TRAIL expression in ASC was about 3.3 times higher than in macrophages, suggesting that ASCs are the major TRAIL source.”

Additionally, although the TRAIL expression was 9.1-fold higher in ASCs co-cultured with macrophages than in ASCs cultured at high-density, the conditioned medium (CM) we used in our experiments (Figure 1C, 2B, and 2C) was obtained from high-density cultured ASCs, and not from the co-culture condition (and consequently, the effects we report are ASCs-TRAIL-mediated). The main reason for our rationale was that various cytokines secreted by macrophages could make our results difficult to interpret. Still, our approach also excluded the effect of macrophage-derived TRAILs derived from macrophages. Therefore, we do not think it is necessary to add a protocol describing how to avoid contamination of macrophage-derived substances (e.g. mRNA or proteins).

2 the experimental design in figure 2 is not rigorous and compromised by the possibility that other components than TRAIL in the CM may modulate the corresponding phenotypes on tumor cells. A good design is using TRAIL KO ASCs as a control group side by side with the CM from the WT ASCs. Meantime, the reviewer is curious about the CM used in figure 2. The authors claimed that the CM was derived from high-density cultures of ASCs, which may be greatly different with that from macrophages cocultured ASCs. If the authors still want to claim the role of M1 macrophages in inducing TRAIL and the consequence on the tumor cells, CMs from M1 macrophages cocultured WT and TRAIL KO ASCs should be also investigated.

Response: We greatly appreciate this critical comment from the reviewer and agree with the reviewer’s interpretation. In this study, we noted that TRAIL secreted by ASCs cultured at high-density can regulate CD133+ cancer stem cells (CSCs) and macrophages can increase TRAIL expression in ASCs. Still, and as the reviewer may know, various inflammatory cytokines secreted by macrophages were reported to control the initiation, progression, and metastasis of tumors. We agree that would be important to explore the effects of CM from WT-ASCs and TRAILKO-ASCs, as described by the reviewer. However, to do so in the context of co-cultures with macrophages might lead to less clear results. The various cytokines derived from macrophages could influence the regulation of CD133+ CSCs by the TRAIL expressing ASCs. Therefore, at least in this study, to disclose the mechanism of action of TRAIL, we decided not to use CM from co-cultures.

In conclusion, we conducted this study as a preliminary approach to understand if the tumor suppression effect of stem cells could be amplified when ASCs act together with macrophages. In the future, as the reviewer commented, we want to investigate the anticancer effects of CM from WT-ASCs and TRAILKO-ASCs co-cultured with M1 macrophages. In addition, we plan to analyze the interaction of stem cells, macrophages, and CD133 CSCs and their possible correlation with the stages of AOM/DSS colon cancer model.

3 the function of TRAIL in CD133+ tumor cells is not novel. Therefore, to further support the authors’ claim in Figure 3, CMs from both TRAIL expressing ASCs should be tested in both WT and CD133 KO LoVo cells.

Response: We greatly appreciate this critical comment from the reviewer. As per your advice, we investigated the cytotoxicity of CM obtained from ASCs cultured at high-density in LoVo and LoVo-CD133 KO and added the results to Figure 3C. Furthermore, we explained these new results in the sub-section 2.3 of the manuscript, as follows (lines 139-141) of the revised manuscript):

“In addition, CM obtained from ASCs cultured at high-density increased apoptosis of LoVo cells, while no apparent effect was detected for LoVo-CD133 KO cells (Fig. 3C).”

4 Again, TRAIL KO ASCs should also be studied in vivo to further support their claims in Figure 4-6. Also, I do not see any evidence here regarding the in vivo role of M1 macrophages. Do the authors just want to simply show that M1 macrophages promote TRAIL secretion from ASCs in vitro?

Response: We greatly appreciate this critical comment from the reviewer. Mesenchymal stem cells are very difficult to genetic-engineer using conventional methods such as lipofection, magnetofection, or electrophoration. Furthermore, the proliferation rate of ASCs is much slower than the one in cancer cells, and it is reported that aging is induced in the process of building stable cells. Still, we are trying to overcome all these limitations to generate genetically-engineered ASCs overexpressing TRAIL, so we can analyze their effect in the suppression of colon cancer, as a next step. We further plan to use these cells in macrophage-deficient mice, in the context of cancer development, to completely understand the role of macrophages. This was our plan for a future study, that now, following your advice, will also contemplate TRAIL-deficient ASCs.

Minor comments

1 It is surprising that CD133 KO LOVO cells exhibit slightly reduced apoptosis compared to the control cells in Figure 3C, which contradicts the tumor-promoting role of CD133. The authors should re-check this data. In addition, Figure 3B is kind of confusing to me. Do the authors suggest that CD133 KO does not affect the cell viability of LOVO cells?

Response: We greatly appreciate this critical comment from the reviewer. To better show that LoVO-CD133 KO cells show resistance to TRAIL (compared to TRAIL-sensitive LoVo cells), we plotted the relative cytotoxicity of each cell before and after TRAIL treatment in Fig. 3. However, we understand that this interpretation can be confusing to reviewers and readers. Therefore, we have also have replotted TRAIL-induced cytotoxicity in LoVo and LoVo-CD133 KO cells using absolute absorbance values.

2 the authors should describe how they prepare M1 THP-1 macrophages. This is important to interpret the data regarding the specific role of M1 macrophages in Figure 1. It is also interesting to test if M2 macrophages show similar or different behavior in promoting the secretion of TRAIL from ASCs. 

Response: We greatly appreciate this important comment note, and apologize for not including this important protocol in the first submission of our manuscript. Now, as per your advice, and to allow the potential complete reproduction of our work, we describe the conditions used to culture THP-1 and to differentiate them into M1-macrophages in the section of 5.1 (cell culture) of the revised manuscript (lines 305-311). The introduced info reads:

The human monocytic cell line THP-1 was maintained in complete RPMI-1640 (Gibco; supplemented with 10% FBS, penicillin/streptomycin, and 2 mM L-glutamine). Macrophage differentiation (from THP-1 cells) was induced with 100 nM of phorbol ester 12-O-tetradecanoylphorbol-13-acetate (TPA, Sigma) for 2 days. Macrophages were co-cultured with ASCs (in trans-well plates; Corning, Lowell, MA, USA) under treatment with 20 ng/ml of IFN-γ (R&D Systems, Minneapolis, MN, USA) and 10 pg/ml of lipopolysaccharide (LPS, Sigma). After indirect co-cultures, total mRNA and proteins were recovered separately from macrophages and ASCs.

Round 2

Reviewer 1 Report

I accept this modified manuscript.

Reviewer 2 Report

Though the story here is interesting, both the design and the data are neither solid nor able to justify the authors’ claim.  Authors have done some efforts to address my concerns but left most comments, which are also significant questions for evaluation of the scientific value of this manuscript, untouched. I would recommend to only reconsider this manuscript until a comprehensive revision is made.